# Immunotherapy for Triple-Negative Breast Cancer: Combination Strategies to Improve Outcome

**DOI:** 10.3390/cancers15010321

**Published:** 2023-01-03

**Authors:** Liying Li, Fan Zhang, Zhenyu Liu, Zhimin Fan

**Affiliations:** Department of Breast Surgery, General Surgery Centre, The First Hospital of Jilin University, Changchun 130012, China

**Keywords:** triple-negative breast cancer, immunotherapy, immune checkpoint, tumor immune microenvironment, clinical trials

## Abstract

**Simple Summary:**

For decades, countless efforts have been devoted to developing targeted drugs to improve the prognosis of triple-negative breast cancer (TNBC). Among the novel therapies that have been approved for the clinical management of TNBC, immunotherapy shows great potential. Although exciting progress has been made in immunotherapy for TNBC, there are still gaps to fill. This review will analyze current immunotherapy strategies in TNBC, summarize the current landscape of clinical trials, review the results achieved, and shed light on future developments.

**Abstract:**

Due to the absence of hormone receptor (*both estrogen receptors and progesterone receptors*) along with human epidermal growth factor receptor 2 (HER-2) amplification, the treatment of triple-negative breast cancer (TNBC) cannot benefit from endocrine or anti-HER-2 therapy. For a long time, chemotherapy was the only systemic treatment for TNBC. Due to the lack of effective treatment options, the prognosis for TNBC is extremely poor. The successful application of immune checkpoint inhibitors (ICIs) launched the era of immunotherapy in TNBC. However, the current findings show modest efficacy of programmed cell death- (ligand) 1 (PD-(L)1) inhibitors monotherapy and only a small proportion of patients can benefit from this approach. Based on the basic principles of immunotherapy and the characteristics of the tumor immune microenvironment (TIME) in TNBC, immune combination therapy is expected to further enhance the efficacy and expand the beneficiary population of patients. Given the diversity of drugs that can be combined, it is important to select effective biomarkers to identify the target population. Moreover, the side effects associated with the combination of multiple drugs should also be considered.

## 1. Introduction

Triple-negative breast cancer (TNBC) accounts for approximately 15–20% of breast malignancies and is the only subtype of breast cancer that lacks targeted treatment [1]. Compared with other subtypes, TNBC is more aggressive, and most patients develop recurrence and metastasis within 3 years, with poor prognosis [2]. Anthracycline- and taxane-based chemotherapy remains the mainstay of treatment for early-stage patients, but resistance has emerged [3]; for patients with recurrence or metastasis, there are even fewer treatment options. There is an urgent need for novel and more effective treatments.

Genomic advances reveal a high degree of heterogeneity in TNBC and set the stage for the development of targeted therapies [4,5,6]. In the past few years, poly ADP-ribose polymerase inhibitors (PARPi), programmed cell death- (ligand) 1 (PD-(L)1) inhibitors, and antibody–drug conjugates (ADCs) have been approved successively for the treatment of TNBC. Among these, PD-1 inhibitor pembrolizumab is approved for patients with advanced PD-L1-positive and early high-risk disease, displaying great therapeutic potential.

Some studies have found that TNBC has higher PD-L1 expression and tumor-infiltrating lymphocytes (TILs) in comparison with other subtypes [7,8], making it the most likely subtype to benefit from immunotherapy. However, PD-(L)1 inhibitors benefit only a small proportion of individuals, with a single-agent effectiveness of approximately 20% in patients with PD-L1-positive metastatic TNBC (mTNBC) [9]. Numerous preclinical and clinical studies have been conducted to investigate the reasons for this disappointing result, expand the beneficiary population, and improve the efficacy of immune checkpoint inhibitors (ICIs).

In this review, we will analyze the reasons for the poor efficacy of PD-(L)1 inhibitors monotherapy in terms of tumor immune escape mechanisms and tumor immune microenvironment (TIME) characteristics of TNBC. We summarize the current landscape of clinical trials in TNBC, highlight the major immune combination therapy strategies in clinical practice and the progress achieved, and briefly discuss the biomarkers for predicting immunological response, as well as possible adverse events associated with immunotherapy.

## 2. Rationale of Immunotherapy and the TIME of TNBC

The cross-talk between tumor cells and the TIME can be described as “cancer immunoediting”, encompassing three stages: (1) elimination; (2) equilibrium, in which the host immune system and tumor cells that survive enter a dynamic balance; and (3) escape [10] (Figure 1).

Normally, a cascade of events needs to be initiated for the elimination of tumor cells. First, tumor cells release specific antigens that are captured and processed by antigen-presenting cells (APCs), which mainly include dendritic cells (DCs). Next, DCs migrate to lymphoid tissue to present antigenic signals to T cells. Later, T cells initiate, activate, and then transport and infiltrate tumor tissues. Finally, T cells specifically recognize and kill tumor cells [11]. B cells and innate immune cells, such as natural killer (NK) cells, are also essential for the elimination of tumor cells (Figure 1).

However, tumors can evade surveillance and attack from the immune system via complex intrinsic signaling or external microenvironment mechanisms. In general, this can be summed up in these aspects. First, immunogenicity of tumors is decreased via downregulation of antigen expression, sequestering of antigens, or downregulation of major histocompatibility complex (MHC) molecules [12,13]. Second, there are functional and/or quantitative defects in intrinsic and/or adaptive immune cells, including recruitment failure, insufficient maturation, failure to activate, and impaired chemotaxis and transport [14]. Further, an immunosuppressive microenvironment is generated through an increase in the proportion of immunosuppressive cells (regulatory T cells (Tregs), M2 macrophage cells, and myeloid-derived suppressor cells (MDSCs)) and cytokines, as well as the accumulation of immunosuppressive substances [15,16]. Moreover, there is an upregulation of immune checkpoints, a negative regulatory mechanism used by the body to prevent over-activation of the immune system and to protect normal tissues from the autoimmune system [17,18] (Figure 1).

Based on the above information, immunotherapy has emerged that involves the use of various agents or means to enhance immune system function (in a tumor-localized rather than whole-body manner) or to block immune escape pathways of tumor cells (by normalizing anti-tumor immunity), thereby recognizing and eliminating tumor cells. These immunotherapy approaches include: therapeutic cancer vaccines (TCVs), which are active immunotherapies; targeted monoclonal antibodies (mAb) and their derivatives, such as ADCs, adoptive cell therapies (ACTs), and cytokines, which are passive therapies; as well as the best-known ICIs [19].

Tumors that lack immune cell infiltration and do not respond to ICIs are referred to as “cold tumors”, while tumors with high levels of immune cell infiltration and upregulated immune checkpoints that may respond to ICIs are referred as “hot tumors” [20]. Based on omics analysis, heterogeneity of the TIME in TNBC has been revealed. According to the data from these studies, the percentage of TNBC cases that present with “hot tumors” is approximately 25%, which may explain the poor clinical efficacy of ICIs as single agents [21,22,23]. Moreover, the suppressive TIME found in “hot tumors” and the decreased MHC of tumor cells also make ICIs less effective [14].

When all of these findings are considered, it is clear that combining ICIs with other therapies that block immune escape pathways of tumor cells or convert “cold tumors” to “hot tumors” is a potential strategy to improve the clinical response of ICIs.

## 3. The Landscape of Clinical Trials on TNBC Immunotherapy

As of 1 September 2022, we had screened a total of 234 clinical trials on *clinicaltrials.gov* (accessed on 1 September 2022), that primarily explore immunotherapy for TNBC, either as a monotherapy or in combination with other therapies (Figure 2). Apart from that, more than 100 immunotherapy clinical trials are being conducted on advanced solid tumors, including TNBC. Of note, these are just the tip of the iceberg as there are trials registered on other websites. In addition to observing the surge in clinical trials of immunotherapy for TNBC in the past decade, we further analyzed the data and found that: (1) there are 197 trials involving ICIs that target PD-(L)1, accounting for 84% of all trials, of which only 13 trials involved monotherapy, while in the other trials, PD-(L)1 inhibitors were combined with almost all therapies that can be applied to TNBC; (2) there are considerable numbers of clinical trials in both early and metastatic settings, which demonstrates the generalizability of immunotherapy for TNBC; and (3) there is a gradual increase in the number of window-of-opportunity (WOP) trials to test for optimal timing of interventions and changes in biomarkers.

## 4. Performance of PD-(L)1 Inhibitors Monotherapy

PD-(L)1 is the most powerful immune checkpoint found in TNBC. PD-1 can be expressed on a variety of immune cells, the most important of which are activated T cells. PD-1 has two ligands: PD-L1 and PD-L2 [24]. PD-L1 is abnormally overexpressed on the surface of tumor cells and some APCs [25]. Through binding to PD-1, it leads to lymphocyte apoptosis, unresponsiveness, and abnormal secretion of cytokines, thus mediating immune escape of tumor cells [25]. In contrast, PD-L2 has a dual role of suppressing and activating T cells, and its role in tumors is gaining attention [26]. PD-(L)1 inhibitors are monoclonal antibodies that block the PD-1/PD-L1 signaling pathway and restore the immune function of T cells to kill tumor cells.

### 4.1. In Advanced TNBC

The phase Ib study, KEYNOTE-012, first demonstrated acceptable safety and durable anti-tumor activity of single-agent pembrolizumab in previously treated patients with advanced PD-L1-positive TNBC. The objective response rate (ORR) in 32 TNBC patients was 18.5% [27]. The phase II KEYNOTE-086 study then further confirmed the safety and anti-tumor activity of pembrolizumab as first-line or second-line and beyond therapy in patients with mTNBC, using more appropriate dosages and dosing intervals. This study found that pembrolizumab monotherapy performed better in previously untreated PD-L1-positive patients with an ORR of 21.4%, whereas the response was flat in heavily pre-treated mTNBC patients with an ORR of 5.3%. Although there was no significant difference in ORR, progression-free survival (PFS), and overall survival (OS) between the PD-L1-positive and PD-L1-negative subgroups in pre-treated mTNBC, a more durable clinical response was observed in PD-L1-positive patients [28,29]. In the next large phase III randomized KEYNOTE-119 trial, as second- or third-line treatment for mTNBC, pembrolizumab failed to significantly improve OS compared to chemotherapy in the total population (9.9 vs. 10.8 months, HR = 0.97, 95% CI: 0.82 to 1.15) or in the PD-L1-positive population (10.7 vs. 10.2 months, HR = 0.86, 95% CI: 0.69 to 1.06, *p* = 0.073). However, the improved effect of pembrolizumab was consistent with increased tumor PD-L1 expression in the efficacy endpoints of ORR, PFS, and OS [30] (Table 1).

In addition to the above, some trials also tested single-drug ICIs targeting PD-L1 in a metastatic setting. In the phase Ib JAVELIN trial, avelumab showed acceptable safety and clinical activity in the mTNBC subgroup. The ORRs were 5.2% and 22.2% for the total TNBC population and PD-L1-positive TNBC patients, respectively [31]. Moreover, a phase Ia study (NCT01375842) verified that atezolizumab monotherapy was well tolerated in patients with mTNBC and the ORR for the unselected TNBC population was 10%. The PD-L1 expression status and prior treatment history continue to strongly influence the efficacy of atezolizumab [32]. In addition, durvalumab was tested in the SAFIR02-BREAST IMMUNO trial as a maintenance therapy. Subgroup analysis showed that, compared with maintenance chemotherapy, durvalumab improved OS in patients with mTNBC (21.2 vs. 14 months, HR = 0.54, 95% CI: 0.30 to 0.97, *p* = 0.0377), especially in PD-L1-positive patients (27.3 vs. 12.1 months, HR = 0.37, 95% CI: 0.12 to 1.13, *p* = 0.0678) [33] (Table 1).

The preliminary results of these trials suggest that PD-(L)1 blockade alone has a modest clinical response across the entire mTNBC population. However, more durable responses have been observed in specific patients, such as PD-L1-positive patients receiving first-line treatment. These findings encourage further research on PD-(L)1 inhibitors.

### 4.2. In Early-Stage TNBC

There are studies on PD-(L)1 inhibitors monotherapy currently underway in early-stage TNBC patients. SWOG 1418 is an ongoing phase III trial investigating the efficacy of pembrolizumab on TNBC patients with ≥ 1 cm residual invasive cancer or positive lymph nodes after neoadjuvant chemotherapy (NACT) [34]. One year of postoperative intravenous avelumab is currently being evaluated for its impact on survival in high-risk TNBC patients in the A-Brave trial [35]. The results of these trials are eagerly anticipated and could provide additional options for the intensive treatment of patients with early-stage, high-risk TNBC (Table 1).

## 5. Research Progress of PD-(L)1 Inhibitors in Combination with Chemotherapy

Preclinical and clinical studies have shown that in addition to direct toxicity to tumor cells, some chemotherapeutic agents kill tumor cells through a pathway of immunogenic cell death (ICD), which stimulates the recruitment and maturation of APCs, enhances the antigen presentation process, and promotes the activation of T cells [36]. Chemotherapeutic agents also increase the immunogenicity of tumors by exposing MHC molecules and antigens on the surface of tumor cells [37]. The transient immunosuppression induced by chemotherapy causes a massive release of cytokines and chemokines, which increases the infiltration and activation of immune cells [38]. Furthermore, chemotherapeutic agents reduce immunosuppressive cells such as Tregs and MDSCs [39]. These chemotherapy drugs include anthracyclines, cyclophosphamide, and others commonly used for TNBC. Thus, combining PD-(L)1 inhibitors with chemotherapy is a promising approach to enhance the efficacy of immunotherapy and facilitate synergistic anti-tumor activity. Based on this concept, a number of trials combining chemotherapy and immunotherapy are being conducted in the clinic and some breakthroughs have been made (Table 2).

### 5.1. In Advanced TNBC

At present, a majority of clinical trials apply immunotherapy with chemotherapy concomitantly. The reason for this is that ICIs take time to work, while chemotherapy agents kill tumor cells and modify the TIME during this waiting period.

KEYNOTE-355 is a phase III randomized controlled trial (RCT) assessing the efficacy and safety of pembrolizumab plus chemotherapy versus placebo plus chemotherapy as first-line treatment for patients with advanced TNBC; chemotherapy regimens were based on the physician’s choice including nab-paclitaxel, paclitaxel, and gemcitabine/carboplatin. Initial results showed that the combination of pembrolizumab with chemotherapy improved PFS in the intention-to-treat (ITT) population and in the combined positive score (CPS) ≥ 1 subgroup (7.5 vs. 5.6 months, HR = 0.82, 95% CI: 0.69 to 0.97 and 7.6 vs. 5.6 months, HR = 0.74, 95% CI: 0.61 to 0.90, *p* = 0.0014, respectively); the improvement was particularly significant in the CPS ≥ 10 subgroup (9.7 vs. 5.6 months, HR = 0.65, 95% CI: 0.49 to 0.86, *p* = 0.0012) [40]. According to the latest release of follow-up data, OS was improved by almost 7 months in the CPS ≥ 10 subgroup after the addition of pembrolizumab to chemotherapy (23.0 vs. 16.1 months, HR = 0.73, 95% CI: 0.55 to 0.95, *p* = 0.0185) and the adverse effects were manageable [41].

Another single-arm phase Ib/II trial, KEYNOTE-150, used pembrolizumab in combination with eribulin mesylate in patients with mTNBC who had received ≤ 2 lines of prior therapy in the metastatic setting. Of the 167 patients enrolled, 40% had not received previous systemic therapies and were classified in stratum 1. The results showed that the survival benefit was most significant in PD-L1-positive patients who had not received prior systemic therapy, which was consistent with previous studies. This study offers a new immuno–chemotherapy combination for the treatment of patients with mTNBC, although further confirmation is needed [42].

After a phase Ib trial (NCT01633970) demonstrated the safety and feasibility of atezolizumab plus nab-paclitaxel in patients with locally recurrent or metastatic TNBC [43], the efficacy of this immuno–chemotherapy combination for TNBC patients who did not receive systemic therapy in the metastatic setting was further validated by IMpassion130, the first phase III RCT of immunotherapy for TNBC [44]. Preliminary results showed that a PFS benefit was observed with the addition of atezolizumab in both the ITT population (7.2 vs. 5.5 months, HR = 0.80, 95% CI: 0.69 to 0.92, *p* = 0.002) and the PD-L1-positive population (7.5 vs. 5.0 months, HR = 0.62, 95% CI: 0.49 to 0.78, *p <* 0.001). The second set of interim results indicated that atezolizumab significantly improved OS from 18.0 months to 25.0 months in the PD-L1-positive subgroup (HR = 0.71, 95% CI: 0.54 to 0.94), but the difference was not significant in the ITT population (21.0 vs. 18.7 months, HR = 0.86, 95% CI: 0.72 to 1.02, *p* = 0.078) [45]. As a result, in March 2019, atezolizumab was granted accelerated approval by the Food and Drug Administration (FDA) to be used in combination with nab-paclitaxel as a first-line treatment for late-stage TNBC patients. Additionally, the 7.5-month survival benefit shown in the final OS data further demonstrated the durable efficacy of this treatment combination for PD-L1-positive patients (25.4 vs. 17.9 months, HR = 0.67, 95% CI: 0.53 to 0.86) [46].

However, these findings were contradicted when atezolizumab was combined with paclitaxel and compared to placebo plus paclitaxel in the phase III clinical study IMpassion131, which also investigated this as first-line treatment for patients with advanced or metastatic TNBC. The study found no obvious differences in PFS between the two arms, regardless of PD-L1 expression status (5.7 vs. 5.6 months, HR = 0.86, 95% CI: 0.70 to 1.05 for the ITT population and 6.0 vs. 5.7 months, HR = 0.82, 95% CI: 0.60 to 1.12, *p* = 0.20 for the PD-L1-positive patients). With respect to OS, the atezolizumab arm appeared to be worse but not detrimental in the PD-L1-positive population (22.1 vs. 28.3 months, HR = 1.11, 95% CI: 0.76 to 1.64) and in the ITT population (19.2 vs. 22.8 months, HR = 1.12, 95% CI: 0.88 to 1.43). Different chemotherapeutic agents, steroid pre-treatment with paclitaxel, and subtle differences between study populations may explain the difference in results between IMpassion130 and IMpassion131 [47]. In addition, levels of TILs, breast cancer susceptibility gene (BRCA) mutational load, and the proportion of patients with residual disease after NACT (which were unreported in the trial) may also have contributed to the unclear results from IMpassion131 [48]. As the reason for this discrepancy remains undefined, Roche has voluntarily withdrawn the indication for atezolizumab for the treatment of PD-L1 positive advanced TNBC. Recently, a small sample-based single-cell sequencing study suggested that paclitaxel may affect the efficacy of atezolizumab by reducing key anti-tumor immune cells in the TIME but enhancing immunosuppressive macrophages, yet this finding needs to be further explored [49].

Despite some setbacks in the exploration of combination treatments with taxanes, atezolizumab is still being tested in different trials to investigate the safety and efficacy of combination treatment with other chemotherapy agents in TNBC (Table 2).

Beyond concurrent chemotherapy, the induction use of small doses of chemotherapeutic agents prior to immunotherapy is another strategy of the immuno–chemotherapy combination that is in the experimental phase. In the five cohorts of the phase II TONIC trial, patients with mTNBC received no induction or 2 weeks induction with low-dose cyclophosphamide, cisplatin, doxorubicin, and hypofractionated irradiation, respectively, all followed by the PD-1 blocking drug nivolumab. The total ORR was 20%, with a median PFS (mPFS) of 1.9 months; a higher ORR occurred in the doxorubicin and carboplatin cohorts at 35% and 23%, respectively. Analysis of patient samples suggested that short-term doxorubicin or cisplatin induction can convert the tumor microenvironment towards inflammation and improve the response of nivolumab in TNBC. However, due to the limitations of the trial itself, this conclusion needs further confirmation [50]. The subsequent trial, TONIC-2, is currently recruiting (Table 2).

### 5.2. In Early-Stage TNBC

Studies based on transcriptomics and immunohistochemical techniques have revealed that mTNBC has significantly reduced expression of immune activation genes as well as immunotherapeutic targets, such as PD-L1, and a lower number of TILs compared to primary TNBC [51,52]. Thus, the TIME in the early-stage disease setting is more suitable for ICIs to function and to potentially achieve a true cure.

Several trials have indicated that the combination of pembrolizumab with chemotherapy can improve pathological complete remission (pCR) rates in early-stage TNBC. One of the cohorts in the I-SPY2 trial first determined the feasibility of 4 cycles of pembrolizumab in combination with paclitaxel- and anthracycline-based chemotherapy regimen in women with early-stage, high-risk HER-2-negative breast cancer. Compared to standard NACT regimens, the addition of pembrolizumab increased the pCR rate for patients with TNBC by 38% (60% vs. 22% for pembrolizumab vs. control) [53]. Another phase Ib KEYNOTE-173 trial with a relatively small sample volume evaluated the safety and efficacy of pembrolizumab in combination with chemotherapy regimens, including different doses of nab-paclitaxel with or without carboplatin followed by doxorubicin and cyclophosphamide; the overall pCR rate was consistent with I-PSY2 at 60% [54].

In the phase III trial KEYNOTE-522, 1174 patients with previously untreated early-stage TNBC were randomized in a 2:1 ratio to either the pembrolizumab–chemotherapy arm or the placebo–chemotherapy arm (chemotherapy backbone of 4 cycles of paclitaxel plus carboplatin, followed by 4 cycles of anthracycline plus cyclophosphamide every 3 weeks), with up to 9 cycles of adjuvant pembrolizumab or placebo after surgery. Preliminary results based on the first 602 patients showed that the addition of pembrolizumab increased the pCR rate by 13.6% compared to the placebo–chemotherapy arm (64.8% vs. 51.2%, 95% CI: 5.4% to 21.8%, *p* < 0.001). This benefit was observed in most subgroups, including PD-L1-negative patients [55]. Based on this undifferentiated benefit, in July 2021, the FDA approved pembrolizumab in combination with chemotherapy as a neoadjuvant treatment for early-stage, high-risk TNBC and for continued use as a single agent in the adjuvant phase. Furthermore, recently updated follow-up data after 39.1 months showed that pembrolizumab treatment for almost 1 year reduced the risk of disease progression by 37% (3-year event-free survival (EFS) of 84.5% vs. 76.8%, HR = 0.63, 95% CI: 0.48 to 0.82, *p* < 0.001). This EFS benefit was independent of PD-L1 expression status, which is consistent with previous results and further demonstrates the long-term effectiveness of the perioperative addition of pembrolizumab. At the time of this analysis, data on OS were immature and further follow-up data are expected [56].

In addition, the phase II NeoPACT trial is also ongoing, combining pembrolizumab with carboplatin and docetaxel as neoadjuvant therapy. The results of this study will demonstrate whether similar pCR rates and survival benefits can be achieved by removing anthracyclines from neoadjuvant chemotherapy regimens in early TNBC.

However, the situation becomes more complicated upon review of the results of clinical trials with PD-L1 inhibitors.

The phase II GeparNuevo study compared the efficacy of receiving durvalumab or placebo every 4 weeks in addition to chemotherapy of nab-paclitaxel sequentially with epirubicin and cyclophosphamide. A total of 174 patients with early TNBC were enrolled. It was noteworthy that 117 patients in this study received an additional, 2-week earlier window treatment of durvalumab or placebo before the start of nab-paclitaxel, and 87% of 158 detected patients were PD-L1 positive. The intensive postoperative treatment regimen for patients enrolled in this trial was based on the physician’s choice. In the window cohort, the pCR rates were statistically increased by the addition of durvalumab (61.0% vs. 41.4%, OR = 2.22, 95% CI: 1.06 to 4.64, *p* = 0.035), but not in the whole study population (53.4% vs. 44.2%, OR = 1.45, 95% CI: 0.80 to 2.63, *p* = 0.224). However, it remains uncertain whether this difference was due to one dose of durvalumab window treatment [57]. Surprisingly, after a median follow-up of 43.7 months, significant improvements in 3-year invasive disease-free survival (iDFS), distant disease-free survival (DDFS), and OS were observed in the durvalumab group, even without the adjuvant durvalumab treatment, which contradicts the pCR results obtained initially (iDFS was 85.6% vs. 77.2%, HR = 0.48, 95% CI: 0.24 to 0.97, *p* = 0.036; DDFS was 91.7% vs. 78.4%, HR = 0.31, 95% CI: 0.13 to 0.74, *p* = 0.005; OS was 95.2% vs. 83.5%, HR = 0.24, 95% CI: 0.08 to 0.72, *p* = 0.006) [58]. More studies are needed to elucidate this result and to explore the timing and sequence of ICIs when combined with chemotherapy to treat early-stage TNBC.

The efficacy of 8 cycles of nab-paclitaxel and carboplatin with or without atezolizumab in early-stage, high-risk TNBC was investigated in the NeoTRIPaPDL1 trial, with 4 cycles of anthracycline regimen chemotherapy administered as adjuvant treatment. The published results thus far have shown that the addition of atezolizumab to the neoadjuvant setting did not significantly increase the pCR rate in the ITT population (48.6% vs. 44.4%, OR = 1.18, 95% CI: 0.74 to 1.89, *p* = 0.48) or in the PD-L1-positive subgroup (59.5% vs. 51.9%). Nevertheless, the primary endpoint of the study, the EFS data, still requires further follow-up [59].

On the contrary, in the Impassion031 trial, a significant increase in pCR rates was observed when atezolizumab was combined with a standard nab-paclitaxel- and doxorubicin-based chemotherapy regimen and applied in the adjuvant phase as a single agent (58% vs. 41%, rate difference 17%, 95% CI: 6% to 27%, *p* = 0.0044). The mature long-term survival follow-up data are not available at present. Similar to the KEYNOTE-522 results, the benefit of pCR was not significantly related to PD-L1 expression status. Of note, platinum agents were removed from the NACT regimen in this trial [55,60].

Furthermore, it is noteworthy that anthracyclines were given preoperatively in both the KEYNOTE-522 and Impassion031 trials, whereas anthracyclines were applied postoperatively in the NeoTRIPaPDL1 trial. This may be one reason why the difference in pCR rates between the two arms in the NeoTRIPaPDL1 trial was not significant.

The safety and efficacy of other combinations of PD-(L)1 inhibitors with chemotherapy drugs are also being tested in clinical trials. Last but not least, trials using ICIs plus chemotherapy in the adjuvant phase of early-stage TNBC are underway and the results are awaited with great interest (Table 2).

## 6. Research Progress of PD-(L)1 Inhibitors in Combination with Radiotherapy

Similar to chemotherapy, radiotherapy (RT) has a dual role of mediating DNA damage-induced tumor cell death and immunomodulation, which can make the TIME more inflammatory and facilitate the role of ICIs [61]. Whereas RT acts locally, the systemic side effects are less severe and well tolerated.

A small single-arm phase II trial (NCT02730130) enrolled 17 unselected patients with mTNBC with a median of 3 lines on prior systemic therapy. They received RT with 3000 centigrays (cGy) in five fractions over 5–7 days and pembrolizumab within 1 to 3 days after the first fraction. The median follow-up was 34.5 weeks, with an ORR of 17.6%, mPFS of 2.6 months, and median OS (mOS) of 8.25 months. Although the 3 patients who experienced complete remission were all PD-L1 positive, the analysis showed that PD-L1 status was not associated with therapeutic effects [62].

Another phase II AZTEC trial enrolled 50 patients who had received less than 2 lines of prior systemic therapy to receive RT combined with atezolizumab. Participants were randomly assigned to 20 Gy stereotactic ablative body RT (SABR) in one fraction or 24 Gy SABR in three fractions to irradiate 1–4 lesions with at least one metastasis left unirradiated. Atezolizumab was initiated within 5 days after the last part of RT. The median follow-up was 17 months, with mPFS of 3.1 months. No difference was observed in mPFS between the two groups. PD-L1 expression status and TIL levels (5%) had little effect on the efficacy [63].

In these studies, the combination of pembrolizumab and RT showed modest but encouraging clinical activity in unselected patients and was well tolerated, offering a new treatment idea for pre-treated patients with advanced TNBC. Additional trials are still being explored (Table 3).

## 7. Research Progress of PD-(L)1 Inhibitors in Combination with Targeted Therapy

### 7.1. Combination with PARPi

PARPi are drugs that block the repair of single-strand DNA damage. These drugs kill tumor cells through synthetic lethal effects that are formed by the accumulation of homologous recombination (HR) repair defects for DNA double-strand breaks due to mutations in BRCA1/2. In addition to direct killing of tumor cells, previous in vitro studies have shown that PARPi can stimulate intrinsic immunity and upregulate interferon (IFN) release by activating the cyclic GMP–AMP synthase-stimulator of interferon genes (cGAS-STING) signaling pathway, further upregulating tumor PD-L1 expression and infiltration of CD8+ T cells [64,65,66]. In short, PARPi have the potential to turn cold tumors into hot tumors and set the stage for the application of PD-(L)1 blockers.

KEYNOTE-162 is a single-arm phase I/II trial evaluating the efficacy and safety of pembrolizumab in combination with niraparib in 55 patients with advanced TNBC. The total ORR was 21%, with ORR of 47% vs. 11% and 32% vs. 8% for the two subgroups, respectively, when considering tumor BRCA mutations as well as PD-L1 status [67]. Remarkably, the mPFS in patients with BRCA mutations was 8.3 months, which was nearly 3 months longer than the mPFS of 5.6 months for olaparib reported in the OlympiAD trial or 5.8 months for talazoparib reported in the TALA trial [68,69].

A cohort in the I-SPY2 trial studied the efficacy of adding durvalumab and olaparib to standard NACT regimens of paclitaxel compared to paclitaxel alone. In the TNBC subgroup analysis, although the addition of durvalumab and olaparib increased the pCR rate in the experimental arm by 20% (47% vs. 27%), by comparison with related trials, the investigators concluded that the contribution from olaparib to the increased pCR rate in the I-SPY2 experimental arm was relatively modest [70]. However, survival data from this experiment have not been published and a more reasonable random grouping should also be considered.

PD-(L)1 blockers combined with PARPi have shown initial efficacy in both advanced and early-stage TNBC patients, with more trials ongoing (Table 3).

### 7.2. Combination with ADCs

ADCs consist of three components: mAb, linker, and cytotoxic payload. In addition to targeting antigen-expressing tumor cells for payload delivery, the mAb mediates antibody-dependent cell-mediated cytotoxicity (ADCC), antibody-dependent cell-mediated phagocytosis (ADCP), and/or complement-dependent cytotoxicity (CDC), as well as the unique bystander killing effect of ADCs to clear tumor cells [71,72]. The cytotoxic payload, apart from directly killing tumor cells, also has immunomodulatory effects, as with the chemotherapeutic agents discussed above [73,74]. Furthermore, payload microtubule inhibitors and topoisomerase inhibitors can directly activate DCs and promote their maturation [75,76]. Therefore, ADCs may create a more conducive TIME for the enhancement of PD-(L)1 inhibitors and work synergistically with PD-(L)1 inhibitors to fight against tumors.

Sacituzumab govitecan, an ADC that targets the tumor cell surface antigen trop2 and has the irinotecan metabolite SN-38 as its payload, has been approved by the FDA for patients with advanced TNBC who have received 2 or more lines treatments. Clinical trials are currently underway to evaluate the potential of sacituzumab govitecan in combination with pembrolizumab as first-line treatment for mTNBC.

Ladiratuzumab vedotin is an ADC that targets the zinc transporter protein LIV-1 with the microtubule inhibitor monomethyl auristatin E (MMAE) as a payload. A phase Ib/II trial (NCT03310957) evaluated the safety and efficacy of its combination with pembrolizumab as first-line treatment for advanced TNBC. The initial 51 patients included showed moderate tolerability and a manageable safety profile. Among the 26 patients evaluable for efficacy, the ORR was 54% [77]. This trial is currently underway and initial results are encouraging.

Several additional trials are testing the safety and efficacy of PD-(L)1 inhibitors in combination with ADCs in both early-stage and advanced TNBC (Table 3).

### 7.3. Combination with Small Molecule Inhibitors

The serine/threonine kinase AKT is a key component of the phosphatidylinositol-3-kinase (PI3K)/AKT and mammalian target of rapamycin (mTOR) signaling pathways. Activation of this pathway and its downstream pathways is associated with the growth, invasion, and drug resistance of a variety of tumors and is cross-linked with multiple signaling pathways, such as the mitogen-activated protein kinase (MAPK) pathway [78]. It has been shown that activation of these two pathways is associated with an increase in immunosuppressive cells and cytokines as well as a decrease in IFNγ, interleukin-2 (IL-2), and tumor necrosis factor α (TNFα) [79,80,81]. Therefore, a simultaneous blockade of these pathways as well as PD-(L)1 would confer a better therapeutic effect. Results of a phase Ib trial (NCT03800836) combining the AKT inhibitors ipatasertib, atezolizumab, and paclitaxel or nab-paclitaxel as a first-line treatment for mTNBC showed an ORR of 54% and mPFS of 7.2 months in 114 patients. Subgroup analysis according to PD-L1 status, PIK3CA/AKT1/phosphatase and tensin homolog (PTEN) alteration status, or taxane backbone showed no consistent trend across endpoints. Treatment was generally tolerable [82]. Cobimetinib is a MAPK/extracellular signal-regulated kinase (MEK) inhibitor. In the phase II COLET trial, a combination regimen of cobimetinib with atezolizumab and paclitaxel or nab-paclitaxel as first-line treatment failed to significantly improve ORRs in mTNBC (34.4% for the paclitaxel cohort and 29.0% for the nab-paclitaxel cohort) [83]. These findings suggest that more effort is still needed in the understanding of classical pathways and in clinical translation.

Abnormal morphological and functional vascularity within solid tumors results in hypoxia of tumor tissue and increased immunosuppressive TIME, as well as reduced and suppressed immune cell infiltration and activity [84,85]. Preclinical studies have shown that anti-vascular therapy increases immune cell infiltration and PD-L1 expression in tumor tissues [86]. Thus, anti-tumor vascular therapy is a potential method to convert cold tumors into hot tumors. A phase II clinical trial (NCT03394287) combined camrelizumab with the vascular endothelial growth factor receptor 2 (VEGFR2) tyrosine kinase inhibitor apatinib in patients with advanced TNBC with fewer than 3 lines of systemic therapy. Of the 40 patients included, 10 were treated intermittently with apatinib and 30 were treated continuously. The ORR in the continuous dosing cohort was 43.3%, while no objective response was observed in the intermittent dosing cohort. This trial demonstrated that the combination of the two drugs is safe and it shows a superior clinical response to single drug application [87].

More trials on the combination of ICIs with small molecule inhibitors are underway (Table 3).

## 8. Exploration of PD-(L)1 Inhibitors in Combination with Other Immunotherapies

### 8.1. Combination with Other ICIs

In addition to PD-(L)1, immune checkpoints such as cytotoxic T-lymphocyte antigen-4 (CTLA-4), lymphocyte-activation gene 3 (LAG-3), and T cell immunoreceptor with Ig and ITIM domains (TIGIT) are also significantly upregulated in TNBC; their expression levels are further boosted by PD-(L)1 blockade, which may mediate acquired resistance to PD-(L)1 blockade [88,89]. Therefore, to further reverse the tumor immunosuppressive microenvironment and overcome PD-(L)1 inhibitor resistance, dual ICIs therapies have been developed.

CTLA-4 is a co-suppressor molecule expressed by T cells. As a homologous receptor of CD28, CTLA-4 can replace CD28 and bind to the B7 ligand on the surface of APCs, preventing the activation and proliferation of T cells [90]. Anti-CTLA-4 antibodies enhance tumor cell killing by blocking the CTLA-4-B7 checkpoint pathway or by selectively depleting Treg cells, although this requires further validation [91]. In a small single-arm study (NCT02536794), durvalumab in combination with tremelimumab demonstrated preliminary efficacy and a tolerable safety profile in 7 patients with mTNBC, with an ORR of 43% [92]. In particular, a combination regimen of KN046, a bispecific antibody that targets both PD-L1 and CTLA-4, with nab-paclitaxel for advanced TNBC showed initially promising results in a phase Ib/II trial (NCT03872791), which may herald the coming of the era of bispecific antibodies [93].

Nevertheless, the idea of combining ICIs for the treatment of TNBC met a waterloo in the SYNERGY trial. CD73, a metabolic immune checkpoint, is an ecto-5′-nucleotidase that is expressed on a wide range of cells and works synergistically with CD39 to convert ATP into adenosine. Adenosine is a potent immunosuppressive molecule that suppresses the function of a wide range of immune cells, especially T cells [94]. CD73 is highly expressed in TNBC and is associated with poor prognosis [95]. The preliminary results at week 24 were presented at the European Society for Medical Oncology (ESMO) 2022 and showed that the addition of the CD73 inhibitor oleclumab to chemotherapy and durvalumab did not improve clinical benefits as a first-line treatment for advanced TNBC (the clinical benefit rates were 42.9% vs. 43.3%, respectively) [96].

The mixed results suggest that the functions and interactions of various immune checkpoints still need to be more thoroughly explored. Clinical trials on the effects of PD-(L)1 inhibitors in combination with other ICIs, such as novel phagocytosis checkpoints, are in full swing (Table 3).

### 8.2. TCVs and PD-(L)1 Inhibitors

The practice of utilizing vaccines against breast tumors predates even ICI uses. However, due to limited efficacy, this approach is not widely applied clinically. Personalized TCVs based on neoantigens may benefit specific patients via injection of tumor neoantigens that were extracted from tumor tissues or human body fluids along with adjuvants. Such therapy amplifies the process of antigen capture and presentation, increases the number of tumor-specific effector T cells, and establishes long-term memory to inhibit tumor recurrence [97,98]. The major types currently in trials include: autologous cells, whole/genetically modified tumor cells, or DCs; cancer antigens, DNA/RNA/peptide vaccines; and tumor cell products, such as exosomes [99].

TCVs and PD-(L)1 inhibitors act in different steps of tumor elimination, and thus combination treatment will synergistically activate the entire immune system. Several clinical trials are currently testing the safety and efficacy of TCVs in combination with PD-(L)1 inhibitors for the treatment of TNBC, in both early and advanced stages (Table 3). Notably, a preclinical study showed that the sequence of PD-(L)1 inhibitors and vaccine combinations is critical to treatment efficacy [100], which deserves special attention when conducting clinical trials.

### 8.3. Oncolytic Virus (OVs) and PD-(L)1 Inhibitors

OVs immunotherapy utilizes natural or modified viruses to selectively infect tumor cells and replicate in large numbers, thereby lysing tumor cells without harming normal cells [101]. In addition to direct killing of tumor cells, OVs also enhance host anti-tumor immunity by mediating ICD, promoting the release of TAAs, increasing the recruitment and maturation of immune cells, and regulating the suppressive TIME, all together rapidly and effectively transforming cold tumors into inflammatory tumors [102]. The efficacy of OVs alone or in combination with therapies such as ICIs has been demonstrated in preclinical tumor models of TNBC [103]. While oncorine (H101) and talimogene laherparepvec (T-VEC) have been approved by the Chinese Food and Drug Administration and the FDA for the treatment of head and neck cancer and melanoma, respectively, clinical studies of OVs in TNBC are still in their infancy with few results published. A phase I trial (NCT03256344) evaluated the safety of intrahepatic injection of T-VEC in combination with intravenous atezolizumab in patients with mTNBC or colorectal cancer, and no dose-limiting toxicity (DLT) was seen in the four TNBC patients who could be evaluated [104]. In addition, scientists have developed chimeric oncolytic poxvirus that can express anti-PD-L1 antibodies and are currently in a phase I trial. Several trials combining PD-(L)1 blockade and OVs are underway (Table 3).

### 8.4. ACT and PD-(L)1 Inhibitors

ACT refers to a therapy in which immune-active cells are isolated from tumor patients, expanded, modified, and characterized in vitro, and then infused back for the purpose of directly killing tumor cells or stimulating an immune response to kill tumor cells. Adoptive TILs and genetically modified T cells expressing modified T cell receptors (TCR-T) or chimeric antigen receptors (CAR-T) are currently the most studied, while therapies such as adoptive NK cells and cytokine-induced killer cells (CIK) are also gaining attention. However, in the field of TNBC, this treatment is still in early phase trials. ACT can directly increase populations of immune killer cells in cold tumors, but its efficacy may be greatly reduced due to the presence of immune checkpoints. Therefore, combining ACT with PD-(L)1 inhibitors is a promising approach to enhance anti-tumor efficacy. A phase Ib/II trial (NCT03387085) first demonstrated a safe and tolerable combination treatment of low-dose chemoradiation, TCV, NK cells therapy, and a PD-L1 inhibitor as third- or greater-line therapy for mTNBC. The ORR was 56% and the disease control rate was 78% in the initial enrollment of 9 patients [105] (Table 3). These preliminary encouraging results provide ideas for additional combination therapies. More outcomes are to be expected.

## 9. Potential Therapeutic Targets for Reversing Cold Tumors

Although considerable clinical trials have been conducted on TNBC patients with some achievements, the mechanisms of tumor immunity are still being explored. Meanwhile, some potential therapeutic targets that can convert cold tumors have been identified.

According to a fundamental study, the mRNA N^6^-methyladenosine (m^6^A)-binding protein YTHDF1 can recognize and bind transcripts encoding lysosomal proteases, which in turn increases the translation of lysosomal histone proteases in DCs, resulting in the impaired presentation of tumor neoantigens and T cell initiation. In addition, the anti-tumor effect of PD-L1 blockade was enhanced in the YTHDF1^-/-^ tumor-bearing mouse models [106]. These suggest that the combination of ICIs and YTHDF1 depletion may be a potential new therapeutic strategy. Research on innate immunity activation by STING agonists are also proceeding in full swing.

Tumor stromal fibrosis is one mechanism by which T cell infiltration is restricted in cold tumors. A recent study showed that discoidin domain receptor 1 (DDR1), a collagen receptor with tyrosine kinase activity, can enhance the collagen fibril alignment and impede immune infiltration through the binding of its extracellular domain (ECD) to collagen. Conversely, ECD-neutralizing antibodies could disrupt this alignment, attenuate immune rejection, and inhibit tumor growth [107]. This study suggests that disruption of tumor stromal fibrosis is one way to convert cold tumors and holds promise to improve anti-tumor efficacy in combination with ICIs in the future.

There are also a growing number of studies focusing on the impact of the host nervous system and commensal microbes on anti-tumor immunity. The effects of sympathetic-β-adrenergic signaling on MDSCs’ survival, expression of immunosuppressive molecules such as arginase-I and PD-L1, and proliferation and function of effector T cells in tumor tissues have been revealed in mouse tumor models. A reduction of this signaling contributed to the conversion of tumors to an immunoreactive tumor microenvironment, and this conversion significantly improved the efficacy of PD-1 ICI [108,109]. A multi-omics analysis of a TNBC cohort showed that genera under Clostridiales and the related metabolite trimethylamine N-oxide (TMAO) were positively associated with the TIME activation and immunotherapy efficacy [110]. Although showing promising prospects for converting cold tumors and improving the efficacy of immunotherapy, these aspects of TNBC have not been studied sufficiently as of now, and more research is required.

## 10. Biomarkers for Predicting Immunological Response

### 10.1. PD-L1 Expression and TILs

The predictive value of PD-L1 expression for the efficacy of PD-(L)1 inhibitors in TNBC has been demonstrated in several trials [27,40,44]. However, there are still limitations related to choosing PD-L1 as a predictive marker. First, PD-L1 expression is spatiotemporally variable. It not only evolves over time with disease progression but also varies by metastatic location, with the highest prevalence of positivity in lymph nodes and the lowest in the liver [111]. Moreover, PD-L1 status was found to be less predictive of efficacy in early-stage TNBC compared to late-stage disease, as discussed previously [55,60]. Second, there are a variety of immunohistochemistry assays for PD-L1 expression status detection, but a lack of standard test methods. The five mainstream assays commonly used from two companies are the 22C3, 28-8, and 73-10 assays on the DAKO AutoStainer Link 48 platform and the SP142 and SP263 assays on the Ventana Benchmark Ultra platform. These assays use different primary antibodies to assess PD-L1 expression in tumor cells and/or tumor-infiltrating immune cells with different scoring criteria and definitions of PD-L1 positivity [112,113]. The 22C3 assay uses a combined positive score (CPS) based on both tumor cells and immune cells (lymphocytes and macrophages) staining to determine PD-L1-positive tumors in mTNBC patients for pembrolizumab, with a cutoff value of 10 in KEYNOTE-355. In contrast, the SP142 assay uses the percentage of stained tumor-infiltrating immune cells (IC) to the tumor area to determine PD-L1-positive tumors for atezolizumab, with a cutoff value of 1% in IMpassion130. Occasionally, in some trials, the percentage of tumor cells (TC) stained is also used to assess PD-L1 expression [27,57]. It is worth noting that the three scoring systems differ significantly in terms of algorithms and the types of cells evaluated. A comparative study analyzed the concordance between different PD-L1 assays and the relationship with patient clinical outcomes. The results showed poor equivalence between the different assays and they were not analytically interchangeable. SP142 assay (≥1% IC) detected the least prevalence of PD-L1 positivity at 46.4% (74.9% for SP263 (≥1% IC) and 80.9% for 22C3 (CPS ≥ 1)), with almost all of these patients captured by the other two assays, and these patients had better clinical outcome improvement with the application of atezolizumab [114]. In addition, tissue fixation methods and subjective factors of pathologists may also affect PD-L1 results [115,116]. Finally, with the advent of some new treatment combinations, some patients who are negative for PD-L1 can also profit from ICIs, since some drugs upregulate PD-L1 expression during treatment, which is unpredictable before therapy. Conclusively, the predictive value of PD-L1 expression for the efficacy of PD-(L)1 blockade is undeniable, but is not a determinant. Therefore, caution should be exercised when making treatment decisions based on PD-L1 status.

TILs are a cell population consisting of T cells, B cells, and NK cells, including both tumor-killing and immunosuppressive cells [117]. As with PD-L1 expression, TILs are also spatiotemporally variable [51,111]. In KEYNOTE-086, higher levels of TILs were associated with better ORR [118]. In a biomarker analysis of KEYNOTE-119, high TIL levels were related to better clinical outcomes with pembrolizumab, but not with chemotherapy. Patients with TNBC and TILs ≥ 5% survived longer with pembrolizumab than with chemotherapy, but this difference was not significant [119]. In contrast, IMpassion130 showed that stromal TIL (sTIL) levels were synergistic with PD-L1 expression; when assessed independent of PD-L1, TILs failed to provide prognostic value [111]. A simple method of section staining is recommended to quantify the extent of TIL infiltration [120]. TILs appear to be a promising biomarker for predicting the efficacy of ICIs at a lower cost, but more evidence is needed. Furthermore, in addition to the numerical level, the composition ratio of cellular components, activation status, and spatial location distribution of TILs are additional important factors that deserve further investigation when exploring the predictive effect of TILs on the efficacy of immunotherapy.

### 10.2. TMB and Microsatellite Instability (MSI)/Mismatch Repair Deficiency (dMMR)

Tumor mutational burden (TMB) is a measurement of the number of nonsynonymous somatic mutations in the genome of tumor cells [121]. When TMB > 10 mutations/Mb, neoantigen production becomes common to tumor cells and can be recognized by TILs [121]. High TMB has been associated with efficacy benefits for ICIs in various tumors [122,123]. Despite being the highest TMB subtype of breast cancer, TNBC still has a low mutational load compared to other tumors such as melanoma. One study showed that the median TMB in breast cancer was 2.63 mut/Mb and only 5% of patients had high TMB (>10 mut/Mb), with metastatic tumors having higher TMB. Of these, the median mutational burden in TNBC was 1.8 mut/Mb [124]. Data from 149 TNBC patients in the GeparNuevo trial showed a median TMB of 1.52 mut/Mb, and continuous TMB independently predicted pCR [125]. Data from 253 patients in the KEYNOTE-119 trial showed a positive correlation between TMB and clinical response to pembrolizumab, but not to chemotherapy [126]. However, in another study, high TMB failed to predict response to ICIs. In the TNBC subgroup, 10 patients with high TMB (>10 mut/Mb) had an ORR of 0, compared to 20.5% in patients with low TMB. The reason for the immaturity of TMB as a predictor for the efficacy of ICIs is mainly due to the fact that antigens generated by tumor mutations may not be immunogenic [127].

In fact, MSI/dMMR is one possible cause of high TMB [128]. Although MSI-high/dMMR has been shown to be associated with immunotherapy efficacy in a variety of tumors and has been approved by the FDA as a biomarker for the application of PD-1 blockers in solid tumors [129,130,131], its frequency is extremely low in TNBC, even in the high-level TIL subtype [132,133]. Based on the available evidence, MSI-high/dMMR is not a practical biomarker for screening TNBC patients who are or are not suitable for immunotherapy.

The aforementioned biomarkers predict PD-(L)1 blockade responses either from the perspective of the TIME or the tumor itself, but none of them are perfect. For now, the combination of several biomarkers to screen suitable patients may be more reliable. The most critical point in selecting immunotherapy-sensitive individuals and giving the most appropriate therapy is to identify the immune deficiency at the tumor site; this is quite difficult, especially in patients with metastases, as many mechanisms of immune escape may exist. However, with further understanding of tumor immune mechanisms, individualized and precise immunotherapy becomes increasingly possible, especially with the influx of novel genomics, single-cell sequencing, and artificial intelligence technologies.

## 11. Pseudoprogression and Immune-Related Adverse Events

The unique biological mechanisms of immunotherapy-fighting tumors enable a long-term or even complete response. However, they also require a long response time, which may lead to the emergence of immune-related patterns of pseudoprogression, hyperprogression, or a mixed response [134,135,136,137]. Pseudoprogression refers to an initial increase in tumor size followed by a decrease of tumor burden, and is associated with immune cell infiltration, edema, or necrosis due to immunotherapy [138]. Pseudoprogression after immunotherapy for TNBC has been reported [139], but incidence rates based on large samples are lacking. Previous data suggest that the incidence of pseudoprogression in solid tumors is less than 10%, which implies that some patients who present with progression after treatment are likely to have true progression [140]. Although the immune-related response criteria (irRC), immune-related RECIST (irRECIST), and immune RECIST (iRECIST) have been published to assist clinicians in evaluating response to immunotherapy, these are not yet widely used in clinical practice [141,142,143]. Therefore, in patients presenting with tumor progression after initial immune-based therapy, clinicians must assess patients’ clinical conditions and toxicity responses thoroughly before carefully deciding on subsequent treatments. This is especially important when immunotherapy is used in combination with other therapies that have tumor-killing effects.

Along with durable anti-tumor activity comes immune-related adverse events (irAEs) that are distinct from the toxicity of conventional chemotherapy. These irAEs vary according to the type of immunotherapy, but there are some common features among ICIs [144]. First, irAEs are mostly organ-specific, occurring mainly in immune-related organs, with rare cases reported involving multiple organ events at the same time [144,145]. Second, while some irAEs occur rapidly, others are regularly delayed, even after treatment [144]. Finally, there is no clear relationship between irAEs and ICI dose [144,146]. These features remind us that irAEs require long-term management that cannot be limited to the period of dosing. Furthermore, the appearance of some toxicities often requires interruptions or even permanent discontinuation of dosing. The most common irAEs in breast cancer patients are rashes, followed by thyroid dysfunction (hypothyroidism > hyperthyroidism), and infusion reactions [147]. Although these irAEs are not usually fatal, they often require high-dose corticosteroid treatment, which may lead to a reduced efficacy of immunotherapy along with additional side effects. On the other hand, patients with permanent endocrine organ damage (such as the thyroid) are required to take therapeutic drugs for the rest of their life and their quality of life is therefore compromised. Notably, the current addition of PD-(L)1 inhibitors to conventional therapies in TNBC patients has already increased the incidence of associated irAEs, although severe incident rates are less than 10% [147,148,149]. Lessons from other tumor types show us that some novel immunotherapies, as well as combination treatments with immunotherapies, can lead to a higher incidence of irAEs and even severe cytokine release syndromes [150,151,152]. Therefore, the introduction of novel immunotherapies and new combination regimens is something that should be given extra attention. Early identification and management of irAEs is extremely important. Of particular consideration is the impact of immunotherapy on fertility, as a significant proportion of patients with TNBC are younger than 40 years of age.

## 12. Conclusions

TNBC is the subtype of breast cancer with the worst prognosis. To date, although several targeted drugs have been approved for the treatment of TNBC, the urgent need for improved survival has not been met. The practice of immunotherapy in TNBC is just beginning to take off. An advantage of the later start in this field is that experience can be learned from other tumor types, both successful and failed. Although some progress has been made with respect to ICIs for TNBC, many challenges remain. Clinical results show that only a small proportion of patients with TNBC actually benefit from immunotherapy. Thus, identifying the target population and expanding the efficacy is a top priority. Overall, combination treatment is the way forward, but the combination treatment mode, sequence, dosage, and duration require further exploration and careful attention should be focused on balancing economics and toxicity. As a growing number of preclinical and clinical studies are conducted in this field, we expect to reach the ultimate goal: to select the most suitable patients for immunotherapy, to give the most appropriate immunotherapy or immune-combination therapy, to accurately assess the efficacy of the treatment, and to achieve optimal therapeutic results with minimal toxic damage.

## Figures and Tables

**Figure 1 cancers-15-00321-f001:**
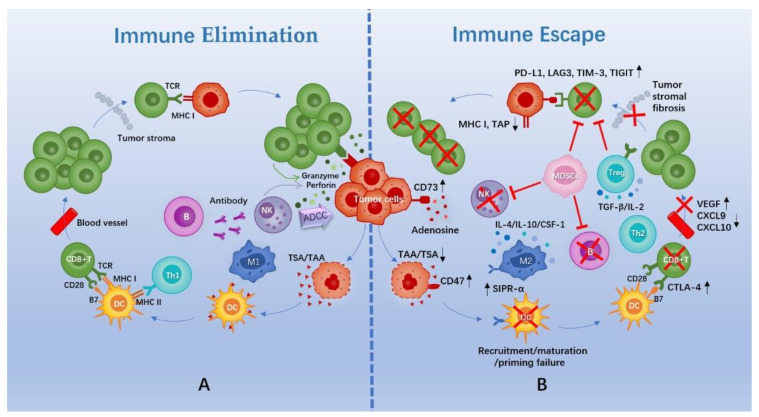
Schematic diagram of immune elimination and escape in TNBC. (**A**) Cascade events of tumor cell elimination by the immune system. (**B**) Mechanisms of tumor cell immune escape. Abbreviations: TAA, tumor-associated antigen; TSA, tumor-specific antigen; DC, dendritic cell; M1/2, M1/2 macrophage; Th1/2, CD4+ helper T cell 1/2; TCR, T cell receptor; MHC, major histocompatibility complex; NK, natural killer cell; ADCC, antibody-dependent cell-medicated cytotoxicity; SIPR-α, signal regulatory protein-α; CTLA-4, cytotoxic T-lymphocyte antigen-4; VEGF, vascular endothelial growth factor; CXCL, chemokine (C-X-C motif) ligand; IL, interleukin; CSF, colony-stimulating factor; TGF, transforming growth factor; Treg, regulatory T cell; MDSCs, myeloid-derived suppressor cells; TAP, transporter associated with antigen processing; PD-L1, programmed death-ligand 1; LAG3, lymphocyte-activation gene 3; TIM-3, T cell immunoglobulin-3; TIGIT, T cell immunoreceptor with Ig and ITIM domains.

**Figure 2 cancers-15-00321-f002:**
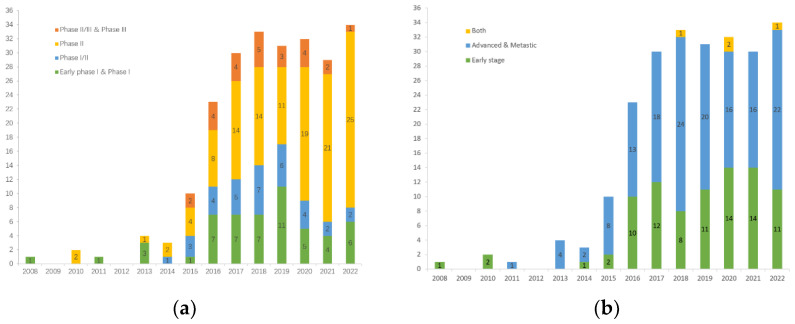
Clinical trials of immunotherapy for TNBC registered at clinicaltrials.gov: (**a**) Number of clinical trials in each phase per year; (**b**) Number of clinical trials by disease stage per year.

**Table 1 cancers-15-00321-t001:** Clinical trials of PD-(L)1 inhibitors monotherapy.

Clinical Trial	Phase	Status	Arms (*n*)	Population (*n*)	PD-L1 Status	Major Outcomes
Trials in advanced TNBC						
KEYNOTE-012 (NCT01848834)	Ib	Completed	Pemb	Pre-treated: PD-L1 (+) (32)	+(stroma/ ≥1% TC ^a^)	ORR: 18.5%
						mPFS: 1.9 months
						mOS: 11.2 months
KEYNOTE-086 (NCT02447003)	II	Completed	Pemb	Cohort A (170): pre-treated	Overall	ORR: 5.3%
						mPFS: 2.0 months
						mOS: 9.0 months
					+(CPS ^b^ ≥ 1)	ORR: 5.7%
						mPFS: 2.0 months
						mOS: 8.8 months
					−	ORR: 4.7%
						mPFS: 1.9 months
						mOS: 9.7 months
				Cohort B (84): pre-untreated, PD-L1 (+)	+(CPS ≥ 1)	ORR: 21.4%
						mPFS: 2.1 months
						mOS: 18.0 months
KEYNOTE-119 (NCT02555657)	III	Completed	Pemb (312) vs. CT ^c^ (310)	Pre-treated: 1–2 prior therapy (622)	Overall	ORR: 9.6 vs. 10.6%
						mPFS: 2.1 vs. 3.3 months
						mOS: 9.9 vs. 10.8 months
					+(CPS ≥ 1)	ORR: 12.3 vs. 9.4%
						mPFS: 2.1 vs. 3.1 months
						mOS: 10.7 vs. 10.2 months
					+(CPS ≥ 10)	ORR: 17.7 vs. 9.2%
						mPFS: 2.1 vs. 3.4 months
						mOS: 12.7 vs. 11.6 months
					+(CPS ≥ 20)	ORR: 26.3 vs. 11.5%
						mPFS: 3.4 vs. 2.4 months
						mOS: 14.9 vs. 12.5 months
JAVELIN (NCT01772004)	Ib	Completed	Avel	Received a median of 2 prior therapies (58)	Overall	ORR: 5.2%
						mPFS: 5.9 months
						mOS: 9.2 months
					+(≥ 10% IC ^d^)	ORR: 22.2%
					−	ORR: 2.6%
NCT01375842	Ia	Completed	Atez	mTNBC: 58% ≥ 2 prior therapies (116)	Overall	ORR: 10%
						mPFS: 1.4 months
						mOS: 8.9 months
					+(≥ 1% IC)	ORR: 12%
						mOS: 10.1 months
					−	ORR: 0%
						mOS: 6.0 months
SAFIR02-BREAST IMMUNO (NCT02299999)	II	Completed	Durv (47) vs. CT (35)	Maintenance setting (82)	Overall	mOS: 21.2 vs. 14 months
					+(≥ 1% IC)	mOS: 27.3 vs. 12.1 months
					−	mOS: 19.5 vs. 14 months
Trials in early-stage TNBC as adjuvant therapy						
SWOG 1418 (NCT02954874)	III	Ongoing	Pemb vs. observation	TNBC with ≥ 1 cm RIC or LN (+) after NACT		NA
A-Brave (NCT02926196)	III	Ongoing	Avel vs. observation	High-risk TNBC		NA

Abbreviations: mTNBC, metastatic triple-negative breast cancer; Pemb, pembrolizumab; Avel, avelumab; Atez, atezolizumab; Durv, durvalumab; NACT, neoadjuvant chemotherapy; CT, chemotherapy; RIC, residual invasive cancer; LN, lymph node; ORR, objective response rate; mOS, median overall survival; mPFS, median progression-free survival; +, PD-L1 positive; −, PD-L1 negative; TC, tumor cells; CPS, combined positive score; IC, immune cells; NA, not available. ^a^ PD-L1 positivity was defined as membranous staining in at least 1% of cells (neoplastic and intercalated mononuclear inflammatory cells) within tumor nests. ^b^ Immunohistochemistry 22C3 assay, CPS = PD-L1 stained cells (including tumor cells, lymphocytes, and macrophages) / the number of all viable tumor cells × 100. ^c^ In KEYNOTE-119, the chemotherapy regimens included capecitabine, eribulin, gemcitabine, or vinorelbine. ^d^ The percentage of PD-L1 stained tumor-associated immune cells in the tumor area; immune cells including lymphocytes, macrophages, dendritic cells, plasma cells, and granulocytes.

**Table 2 cancers-15-00321-t002:** Clinical trials of PD-(L)1 inhibitors in combination with chemotherapy.

Clinical Trial	Phase	Status	Arms (*n*)	Population (*n*)	PD-L1 Status	Major Outcomes
Trials in advanced TNBC						
KEYNOTE-355 (NCT02819518)	III	Ongoing	Pemb + CT ^a^ (566) vs. placebo + CT (281)	First-line treatment in mTNBC (847)	ITT population	mPFS: 7.5 vs. 5.6 months
						mOS: 17.2 vs. 15.5 months
					+(CPS ^b^ ≥ 1)	mPFS: 7.6 vs. 5.6 months
						mOS: 17.6 vs. 16.0 months
					+(CPS ≥ 10)	mPFS: 9.7 vs. 5.6 months
						mOS: 23.0 vs. 16.1 months
KEYNOTE-150/ ENHANCE 1 (NCT02513472)	Ib/II	Completed	Pemb + eribulin mesylate	≤2 prior lines therapies in the metastatic setting (167)	Overall	ORR in total: 23.4%
						stratum 1: 25.8%
						stratum 2: 21.8%
						mPFS in total: 4.1 months
						stratum 1: 4.2 months
						stratum 2: 4.1 months
						mOS in total: 16.1 months
						stratum 1: 17.4 months
						stratum 2: 15.5 months
					+(CPS ≥ 1)	ORR in stratum 1: 34.5%
						ORR in stratum 2: 24.4%
						mPFS in stratum 1: 6.1 months
						mPFS in stratum 2: 4.1 months
						mOS in stratum 1: 21.0 months
						mOS in stratum 2: 14.0 months
					−	ORR in stratum 1: 16.1%
						ORR in stratum 2: 18.2%
						mPFS in stratum 1: 3.5 months
						mPFS in stratum 2: 3.9 months
						mOS in stratum 1: 15.2 months
						mOS in stratum 2: 15.5 months
TORCHLIGHT (NCT04085276)	III	Recruiting	Tori + nab-P vs. placebo + nab-P	≤1 line of CT in the metastatic setting		NA
NCT04537286	II	Recruiting	Cari + nab-P + Cp	First-line treatment in mTNBC		NA
NCT02755272	II	Recruiting	Pemb + Cb + gemcitabine vs. Cb + gemcitabine	>2 prior lines therapies in the metastatic setting		NA
TONIC (NCT02499367)	II	Ongoing	A/C/Cp/ RT/no induction + Nivo (70)	mTNBC (70)		ORR in total: 20%
						Cp induction ORR: 23%
						A induction ORR: 35%
						mPFS in total: 1.9 months
TONIC-2 (NCT04159818)	II	Recruiting	Cp/ low dose A/no induction + Nivo	Metastatic or incurable locally advanced TNBC		NA
NCT01633970	Ib	Completed	Atez + nab-P (33)	≤2 lines prior CT in the metastatic setting (33)		ORR: 39.4%
						mPFS: 5.5 months
						mOS: 14.7 months
IMpassion130 (NCT02425891)	III	Completed	Atez + nab-P (451) vs. placebo + nab-P (451)	First-line treatment in mTNBC (902)	ITT population	mPFS: 7.2 vs. 5.5 months
						mOS: 21.0 vs. 18.7 months
					+(≥1% IC ^c^)	mPFS: 7.5 vs. 5.0 months
						mOS: 25.4 vs. 17.9 months
IMpassion131 (NCT03125902)	III	Ongoing	Atez + P (431) vs. placebo + P (220)	First-line treatment in mTNBC (651)	ITT population	mPFS: 5.7 vs. 5.6 months
						mOS: 19.2 vs. 22.8 months
					+(≥1% IC)	mPFS: 6.0 vs. 5.7 months
						mOS: 22.1 vs. 28.3 months
IMpassion132 (NCT03371017)	III	Recruiting	Atez + CT ^d^ vs. placebo + CT	First-line treatment for locally advanced inoperable or mTNBC		NA
ALICE (NCT03164993)	II	Ongoing	Atez + PLD + C vs. placebo + PLD + C	≤ 1 line previous CT in the metastatic setting		NA
GIM25-CAPT (NCT05266937)	II	Recruiting	Atez + nab-P + Cb	First-line therapy in PD-L1-positive mTNBC		NA
EL1SSAR (NCT04148911)	III	Ongoing	Atez + nab-P	First-line therapy in PD-L1-positive mTNBC		NA
Trials in early-stage TNBC as neoadjuvant therapy						
I-SPY2 (NCT01042379)	II	Recruiting	Pemb + P→AC (29) vs. P→AC (85)	HER-2 negative, stage II or III at high risk (250, including 114 TNBC)		pCR rates in TNBC: 60% vs. 22%
KEYNOTE-173 (NCT02622074)	Ib	Completed	Pemb + (nab-P ± Cb→AC) (60)	High-risk, early-stage TNBC (60)		Overall pCR rate: 60%
KEYNOTE-522 (NCT03036488)	III	Ongoing	Pemb + (PCb→AC/EC) (784) vs. placebo + (PCb→AC/EC) (390) (→surgery→Pemb/placebo for up to 9 cycles)	Stage II-III TNBC (1174)	Overall	pCR rates ^e^: 64.8% vs. 51.2%
						3-year EFS: 84.5% vs. 76.8%
					+(CPS ≥ 1)	pCR rates: 68.9% vs. 54.9%
					−	pCR rates: 45.3% vs. 30.3%
NeoPACT (NCT03639948)	II	Ongoing	Pemb + Cb + docetaxel	Early-stage TNBC		NA
NCT04613674	III	Recruiting	Camr + CT vs. placebo + CT	Early or Locally Advanced TNBC		NA
GeparNuevo (NCT02685059)	II	Completed	Durv×2w ^f^→durv + (nab-P →EC) (88) vs. placebo + (nab-P →EC) (86) (→surgery→physician’s choice)	Primary, cT1b-cT4a-d disease, centrally confirmed TNBC (174)		Overall pCR rates: 53.4% vs. 44.2%
						pCR rates in the window cohort: 61.0% vs. 41.4%
						3-year iDFS: 85.6% vs. 77.2%
						3-year DDFS: 91.7% vs. 78.4%
						3-year OS: 95.2% vs. 83.5%
NeoTRIPaPDL1 (NCT02620280)	III	Ongoing	Atez + nab-P + Cb (138) vs. nab-P + Cb (142) (→surgery→adjuvant anthracycline regimen as per investigator’s choice)	Early high-risk and locally advanced TNBC (280)	ITT population	pCR rates: 48.6% vs. 44.4%
					+(≥1% IC)	pCR rates: 59.5% vs. 51.9%
IMpassion031 (NCT03197935)	III	Ongoing	Atez + (nab-P →AC) (165) vs. placebo + (nab-P →AC) (168) (→surgery→ adjuvant Atez/placebo for up to 11 cycles)	Stage II–III TNBC (333)	Overall	pCR rates: 58% vs. 41%
					+(≥1% IC)	pCR rates: 69% vs. 49%
					−	pCR rates: 48% vs. 34%
NSABP B-59 (NCT03281954)	III	Ongoing	Atez + (PCb→AC) vs. placebo + (PCb →AC) (→surgery→adjuvant Atez/placebo until 1 year after the first dose)	Stage II–III TNBC		NA
NCT02530489	II	Ongoing	Atez + nab-P (→surgery→adjuvant Atez for 4 cycles)	TNBC that were non-responders to initial AC chemotherapy		NA
Trials in early-stage TNBC as adjuvant therapy						
NCT03487666	II	Ongoing	Nivo vs. capecitabine vs. Nivo + capecitabine	TNBC with ≥ 1 cm RIC or LN (+) after NACT		NA
IMpassion030 (NCT03498716)	III	Recruiting	Atez + A/P-based CT vs. CT	Operable-stage II-III TNBC		NA
NCT03756298	II	Recruiting	Atez + capecitabine vs. capecitabine	TNBC with RIC after NACT		NA

Abbreviations: mTNBC, metastatic triple-negative breast cancer; Pemb, pembrolizumab; Atez, atezolizumab; Durv. durvalumab; Cari, carilizumab; Tori, toripalimab; Nivo, nivolumab; Camr, camrelizumab; CT. chemotherapy; RT, radiotherapy; Nab-P, nab-paclitaxel; P, paclitaxel; E, epirubicin; A, doxorubicin; C, cyclophosphamide; Cb, carboplatin; Cp, cisplatin; PLD, pegylated liposomal doxorubicin; mOS, median overall survival; mPFS, median progression-free survival; EFS, event-free survival; iDFS, invasive disease-free survival; DDFS, distant disease-free survival; ITT population, intention-to-treat population; ORR. objective response rate; pCR, pathological complete remission; CPS, combined positive score; IC, immune cells; +, PD-L1 positive; −, PD-L1 negative; RIC, residual invasive cancer; NA, not available. ^a^ In KEYNOTE-355, the chemotherapy regimens included nab-paclitaxel; paclitaxel; or gem citabine plus carboplatin. ^b^ Immunohistochemistry 22C3 assay, CPS = PD-L1 stained cells (including tumor cells, lymphocytes, and macrophages) / the number of all viable tumor cells × 100. ^c^ The percentage of PD-L1 stained tumor-associated immune cells in the tumor area; immune cells including lymphocytes, macrophages, dendritic cells, plasma cells, and granulocytes. ^d^ In the IMpassion132, the chemotherapy regimens include gemcitabine, capecitabine, and car boplatin. ^e^ In KEYNOTE-522, the first interim pCR analysis was conducted on the first 602 patients who underwent randomization (401 patients in pembrolizumab–chemotherapy group and 201 in placebo–chemotherapy group). ^f^ In the GeparNuevo study, 117 patients participated in the window phase.

**Table 3 cancers-15-00321-t003:** Clinical trials of PD-(L)1 inhibitors in combination with radiotherapy, targeted therapy, and other immunotherapies.

Clinical Trial	Phase	Status	Arms	Population
NCT02730130	II	Ongoing	Pemb + RT	mTNBC: a median of 3 lines prior systemic therapy
AZTEC (NCT03464942)	II	Ongoing	Atez + RT	Advanced TNBC: <2 lines of prior systemic therapy
NCT03483012	II	Ongoing	Atez + RT	mTNBC with brain metastases
KEYNOTE-162 (NCT02657889)	I/II	Completed	Pemb + niraparib	Advanced TNBC: a median of 1 prior line of therapy (range, 0–3) in the metastatic setting
I-SPY2 (NCT01042379)	II	Recruiting	Durv + olaparib + paclitaxel vs. paclitaxel	Stage II-III TNBC: preoperative treatment
DORA (NCT03167619)	II	Ongoing	Durv + olaparib	Platinum-treated mTNBC
KEYLYNK-009 (NCT04191135)	II/III	Ongoing	Pemb + olaparib vs. Pemb + Cb + gemcitabine	Locally recurrent inoperable or metastatic TNBC: after induction with first-line CT + Pemb
NCT03594396	I/II	Ongoing	Olaparib + Durv	Stage II/III TNBC or low ER breast cancer: preoperative treatment
NCT03310957	Ib/II	Recruiting	Pemb + ladiratuzumab vedotin	Unresectable locally advanced or metastatic TNBC: first-line treatment
ASCENT-04 (NCT05382286)	III	Recruiting	Pemb + SG vs. pemb + TPC	Previously untreated, locally advanced inoperable, or metastatic PD-L1-positive TNBC
NCT04468061	II	Recruiting	Pemb + SG vs. SG	PD-L1-negative mTNBC
ASPRIA (NCT04434040)	II	Recruiting	Atez + SG	Early-stage TNBC with RIC after NACT
NCT03394287	II	Completed	Camr + apatinib	Advanced TNBC: <3 lines of systemic therapy
NCT05447702	II	Not yet recruiting	Camr + apatinib + CT	Neoadjuvant therapy for stage II-III TNBC
NCT04303741	II	Ongoing	Camr + apatinib + eribulin	Unresectable recurrent or mTNBC; pre-treated with anthracycline and taxane
NCT04427293	I	Recruiting	Pemb + Lenvatinib	Early-stage TNBC: preoperative treatment
NCT04335006	III	Recruiting	Care + nab-P + apatinib vs. Care + nab-P vs. nab-P	Locally advanced or metastatic TNBC: first-line treatment
NCT03800836	Ib	Completed	Atez + ipatasertib + P/nab-P	mTNBC: first-line treatment
BARBICAN (NCT05498896)	II	Ongoing	Atez + PAC + ipatasertib vs. Atez + PAC	Early-stage TNBC: preoperative treatment
NCT04177108	III	Ongoing	Atez/placebo + ipatasertib/placebo + P	Locally advanced unresectable or metastatic TNBC
COLET (NCT02322814)	II	Completed	Atez + cobimetinib + P (cohorts II)/Atez + cobimetinib + nab-P (cohort III)	First-line treatment for mTNBC
NCT02536794	II	Completed	Durv + tremelimumab	Pre-treated mTNBC
NCT03872791	Ib/II	Ongoing	KN046 vs. KN046 + nab-P	mTNBC
SYNERGY (NCT03616886)	Ib/II	Ongoing	Durv + oleclumab +PCb vs. Durv + PCb	First-line treatment for mTNBC
NCT04584112	Ib	Ongoing	Atez + tiragolumab + CT	First-line treatment for PD-L1 (+) mTNBC
NCT05227664	II	Recruiting	AK117 + P/nab-P vs. AK112 + P/nab-P vs. AK117+AK112 + P/nab-P	First-line treatment for mTNBC
NCT03362060	I	Ongoing	Pemb + PVX-410 vaccine	Pre-treated HLA-A2 (+) mTNBC
NCT02826434	I	Ongoing	Durv + PVX-410	HLA-A2 (+) stage II or III TNBC
NCT03606967	II	Recruiting	CT →Durv + tremelimumab + Vaccine vs. CT →Durv + tremelimumab	First-line treatment for PD-L1-negative mTNBC
NCT03199040	I	Ongoing	Durv + DNA vaccine vs. DNA vaccine	Early-stage TNBC
NSABP FB-14 (NCT04024800)	II	Ongoing	AE37 peptide vaccine + Pemb	Advanced TNBC: ≤ 1 line of systemic therapy
NCT03387085	Ib/II	Ongoing	Combination of multiple treatments	mTNBC: ≥ 2 lines of prior therapy
NCT04445844	II	Recruiting	Retifanlimab + pelareorep	mTNBC: received 1–2 prior lines of systemic therapy
NCT03004183	II	Ongoing	ADV/HSV-tk + RT + Pemb +	Pre-treated mTNBC
NCT03256344	I	Completed	Atez + talimogene laherparepvec	mTNBC with liver metastases
NCT05081492	I	Recruiting	CF33-hNIS-antiPDL1	mTNBC: ≥ 2 prior lines of therapy for metastatic disease
NCT04185311	I	Ongoing	Talimogene laherparepvec + nivolumab + ipilimumab	Localized, palpable HER-2 negative breast cancer

Abbreviations: mTNBC, metastatic triple-negative breast cancer; Pemb, pembrolizumab; Camr, camrelizumab; Atez. atezolizumab; Durv. durvalumab; Nivo, nivolumab; Care, carelizumab; CT, chemotherapy; RT, radiotherapy; Nab-P, nab-paclitaxel; P, paclitaxel; E, epirubicin; A, doxorubicin; C, cyclophosphamide; Cb, carboplatin; mOS, median overall survival; mPFS, median progression-free survival; ORR, objective response rate; pCR, pathological complete remission; DOR, median duration of response; SG, sacituzumab govitecan; TPC, treatment of physician’s choice; HLA, human leukocyte antigen; RIC, residual invasive cancer; NACT, neoadjuvant chemotherapy.

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
