# Peer review of "Immunotherapy for Triple-Negative Breast Cancer: Combination Strategies to Improve Outcome"

_cancers, 2023, doi:10.3390/cancers15010321_

Round 1

Reviewer 1 Report

The review article titled Immunotherapy for Triple-Negative Breast Cancer: Combination Strategies to Improve Outcome by Liying Li et al. is comprehensive and gives a clear picture of the state of the field and how we got there. The quality of writing and presentation are outstanding.

Recommendations
1. A section related to immunotherapy approaches description including therapeutic cancer vaccines, targeted monoclonal antibodies, adoptive cell therapies (ACTs), cytokines, and  ICIs would be an attractive addition to the manuscript for readers who are not familiar with the subject.

2. If possible, unify the major outcomes criteria for all the clinical studies listed.

3. Although the article was based on clinical information, it could include preclinical trials of new therapeutic combinations with immunotherapy that have not yet reached the clinical stage but show the way.

Author Response

Dear reviewer 1,

Greetings!

Manuscript ID: cancers-2063031

Title: Immunotherapy for Triple-Negative Breast Cancer: Combination Strategies to Improve Outcome

Thank you very much for taking the time out of your busy schedule to read and revise my article and your valuable suggestions. You have made comprehensive corrections to the structure and content of my paper. It has played a very important role in improving the quality of my paper.

I have carefully studied your comments and have carefully revised the paper according to the suggestions as follows (marked in red).

Recommendation 1. A section related to immunotherapy approaches description including therapeutic cancer vaccines, targeted monoclonal antibodies, adoptive cell therapies (ACTs), cytokines, and ICIs would be an attractive addition to the manuscript for readers who are not familiar with the subject.

We are very thankful to the reviewer for the recommendation. We have carefully proofread and refined the descriptions of immunotherapeutic approaches in the original article, including therapeutic cancer vaccines, oncolytic virus, targeted monoclonal antibodies and their derivatives ADCs, adoptive cell therapies (ACTs) and ICIs. But these descriptions are not concentrated in one paragraph, instead they are scattered in the corresponding sections. For example, we describe the PD-(L)1 immune checkpoint and its inhibitors in lines 137 to 145, while the other two immune checkpoints, CTLA-4 and CD73, and their inhibitors development are highlighted in subsection 8.1. We hope that this arrangement will make it easier for readers to read. If you have better suggestions, please let we know and we are willing to make further modifications.

Recommendation 2. If possible, unify the major outcomes criteria for all the clinical studies listed.

Thank you for your detailed advice. We have modified some major outcomes criteria for clinical studies listed in the paper and table (mainly Table 2) to make them more intuitive and concise and try to ensure consistency of context. However, differences in primary endpoints, enrolled population, and study focus across clinical trials make it difficult to achieve complete consistency in primary outcome criteria. We hope our modifications have met your expectations.

Recommendation 3. Although the article was based on clinical information, it could include preclinical trials of new therapeutic combinations with immunotherapy that have not yet reached the clinical stage but show the way. 

Thank you again for your suggestions for additional content for the paper. We have added preclinical trials of new treatment combinations with immunotherapies in Section 9 (Lines 618 to 649). Please see the revised version of the paper for details.

In the end, I would like to thank you again for your guidance and for reviewing and revising my paper again. I hope that under your guidance I can complete this excellent paper and sincerely hope that my paper will be published in your journal.

Sincerely,

Zhimin Fan

Professor

Department of Breast Surgery, General Surgery Centre, The First Hospital of Jilin University,

Xinmin Street No. 1, Changchun, Jilin, 130021, China

Tel: +86-15754307448;

E-mail: fanzm@jlu.edu.cn

Reviewer 2 Report

This is a well written and comprehensive review on the various strategies of immunotherapy for TNBC. Despite the promise held by immune checkpoint inhibitors, efficacy has been modest. The limitations of the various approaches are well discussed.

There are few small points to be addressed.

Reference 6 quoted in line 49 seems to be incorrect as this paper describes the genomic and transcriptomic landscape.

The author should explain and comment on the different PD-L-1 scoring systems, TPS, CPS, IC and TC that were mentioned in the tables.

Author Response

Response to Reviewer Comments

Dear reviewer 2,

Greetings!

Manuscript ID: cancers-2063031

Title: Immunotherapy for Triple-Negative Breast Cancer: Combination Strategies to Improve Outcome

Thanks very much for taking the time out of your busy schedule to read and revise my article and your valuable suggestions. You have made comprehensive corrections to the content of my paper. It has played a very important role in improving the quality of my paper.

I have carefully studied your comments and have carefully revised the paper according to the suggestions as follows (marked in red).

Recommendation 1. Reference 6 quoted in line 49 seems to be incorrect as this paper describes the genomic and transcriptomic landscape.

We are very thankful to the reviewer for the careful review. We have corrected this error. The original reference 6 quoted in line 49 has been deleted and the revised reference is labeled reference 9.

Recommendation 2. The author should explain and comment on the different PD-L1 scoring systems, TPS, CPS, IC and TC that were mentioned in the tables.

Thank you for your detailed advice. We have added an explanation for the PD-L1 scoring systems mentioned in the table (table 1 and table 2) in a footnote below and discussed the characteristics of these scoring systems in the following text (located in section 10.1, lines 658-679).

In the end, I would like to thank you again for your guidance and for reviewing and revising my paper again. I hope that under your guidance I can complete this excellent paper and sincerely hope that my paper will be published in your journal.

Sincerely,

Zhimin Fan

Professor

Department of Breast Surgery, General Surgery Centre, The First Hospital of Jilin University,

Xinmin Street No. 1, Changchun, Jilin, 130021, China

Tel: +86-15754307448;

E-mail: fanzm@jlu.edu.cn
